# 1,2-Diarylethanols—A New Class of Compounds That Are Toxic to *E. coli* K12, R2–R4 Strains

**DOI:** 10.3390/ma14041025

**Published:** 2021-02-22

**Authors:** Paweł Kowalczyk, Damian Trzepizur, Mateusz Szymczak, Grzegorz Skiba, Karol Kramkowski, Ryszard Ostaszewski

**Affiliations:** 1Department of Animal Nutrition, The Kielanowski Institute of Animal Physiology and Nutrition, Polish Academy of Sciences, 05-110 Jabłonna, Poland; g.skiba@ifzz.pl; 2Institute of Organic Chemistry PAS, Kasprzaka 44/52, 01-224 Warsaw, Poland; damian.trzepizur@icho.edu.pl (D.T.); ryszard.ostaszewski@icho.edu.pl (R.O.); 3Department of Molecular Virology, Institute of Microbiology, Faculty of Biology, University of Warsaw, Miecznikowa 1, 02-096 Warsaw, Poland; mszymczak@biol.uw.edu.pl; 4Department of Physical Chemistry, Medical University of Bialystok, Kilińskiego 1 Str., 15-089 Bialystok, Poland; kkramk@wp.pl

**Keywords:** oxidative stress, Fpg protein, LPS, 1,2-diaryloethanols

## Abstract

An initial study of 1,2-diarylethanols derivatives as new potential antibacterial drugs candidates was conducted. Particular emphasis was placed on the selection of the structure of 1,2-diarylethanols with the highest biological activity of lipopolysaccharides (LPS) in the model strains of Escherichia coli K12 (without LPS in its structure) and R2–R4 (with different lengths of LPS in its structure). In the presented studies, based on the conducted minimum inhibitory concentration (MIC) and MBC tests, it was demonstrated that the antibacterial (toxic) effect of 1,2-diarylethanols depends on their structure and the length of LPS bacteria in the membrane of specific strains. Moreover, the oxidative damage of bacterial DNA isolated from bacteria after modification with newly synthesized compounds after application of the repair enzyme Fpg glycosylases was analysed. The analysed damage values were compared with modification with appropriate antibiotics; bacterial DNA after the use of kanamycin, streptomycin, ciprofloxacin, bleomycin and cloxicillin. The presented research clearly shows that 1,2-diarylethanol derivatives can be used as potential candidates for substitutes for new drugs, e.g., the analysed antibiotics. Their chemical and biological activity is related to two aromatic groups and the corresponding chemical groups in the structure of the substituent. The observed results are particularly important in the case of increasing bacterial resistance to various drugs and antibiotics, especially in nosocomial infections and neoplasms, and in the era of pandemics caused by microorganisms.

## 1. Introduction

Contamination of the hospital environment by pathogenic microorganisms is one of the important problems of hospital epidemiology. A work published by Boyce in the pages of Infection Control & Hospital Epidemiology presents the role of alcohols as disinfectants in the hospital environment [1]. Applications in this area include ethyl alcohol, isopropyl alcohol and n-propanol. Their main activity is protein denaturation. This process requires the presence of water, and therefore aqueous alcohol solutions have a stronger effect than absolute alcohol [2]. Water solutions of alcohols with a concentration of 60–90% have the strongest biocidal effect. Recently, with the example of *Escherichia coli*, it has been shown that the action of ethyl alcohol is also based on the inhibition of protein synthesis by direct influence on ribosomes and RNA polymerase and inhibition of single-stranded mRNA [2]. The action of alcohols covers the vegetative forms of bacteria, fungi, and enveloped viruses. Alcohols practically have no killing effect on bacterial spores, they have a weak effect on non-enveloped viruses, the effect of ethyl alcohol being stronger in the latter case than that of isopropyl alcohol [3]. Alcohols proved to be useful in superficial disinfection of non-critical and semi-critical equipment (stethoscopes, pagers, cell phones), in wiping the surfaces of ampoules and vials [4]. Wiping the openings of vascular catheters with swabs impregnated with chlorhexidine alcohol solution and the use of ports containing 70% isopropyl alcohol are part of the strategy to reduce the risk of catheter-related infections [5,6]. It is also recommended to rinse the working canals of endoscopes with a 70–90% solution of ethyl or isopropyl alcohol after high-degree disinfection [5,6]. The disadvantages of alcohols are their flammability, lower efficiency in the presence of organic pollutants and quick evaporation. Most of the available preparations require 1–10 min of contact with the surface for full effect. Rapid evaporation makes them ineffective in disinfecting large surfaces on which pathogenic microorganisms can accumulate. Bacteria that accompany us throughout our lives, gaining resistance to drugs and attacking a weakened organism, has become a lethal threat [5,6]. Unfortunately, losing antibiotics and disinfectants as an effective weapon against increasingly resistant strains of bacteria, we must reach for newer and stronger compounds that will be more toxic to microorganisms, and their molecules can be used as sprayers for cleaning surfaces and hands.

Alcohol-based hand sanitizers were introduced widely in hospitals at the beginning of the 21st century to stop the expansion of antibiotic-resistant superbugs that die outside the body from alcohol in 20–30 s [7,8]. The analysed compounds are not only an important building block in organic synthesis, but also constitute the structural elements of biologically active substances. Therefore, the developed methodology may in the future be used in the pharmaceutical or chemical industry for the synthesis of important compounds with biological or pharmacological activity containing aryl groups [9,10].

Among the many natural anti-cancer compounds, combretastatin A-4-stilbenoid isolated from African willow bark, deserves special attention. It has a strong antitumor effect and a similar effect to the analysed 1,2-diarylethanols [11,12,13,14,15,16]. This compound was found to have antioxidant and anti-inflammatory properties [11,16,17,18,19,20,21,22,23,24,25,26,27,28,29,30,31,32,33,34,35,36,37,38,39,40,41]. Recently special attention was paid on combrestatin for its antimicrobial activity [11,16,17,18,19,20,21,22,23,24,25,26,27,28,29,30,31,32,33,34,35,36,37,38,39,40,41]. Due to its biological activity, combretastatin A-4 has become a subject of research by both physicians and synthetic chemists [11,12,13,14,15,16]. It can be used as a reference compound for 1,2-diarylethanols analysed by us. Combrestatin belongs to the class of natural stylbenoids that were isolated from the bark of the African willow *Combretum caffrum* in the 1980s. According to African ethnic medicine, powdered willow bark has healing properties and is still very popular today. At the end of the 20th century, scientists managed to determine what was responsible for the medical effects of this drug. These are compounds from the combretastatin group, mainly: combreatastatin A-1, and combretastatin A-4 (CA-4) [11,12,13,14,15,16]. Combretastatin and their derivatives show anti-cancer activity [11,12,13,14,15,16]. They bind to the protein responsible for building the cytoskeleton of cells, tubulin, which prevents the formation of microtubules, and thus no new cells are formed. Combretastatins are also anti-angiogenic compounds—they impair the blood supply to vigorously developing cancer cells [11,12,13,14,15,16]. If there is no blood supply to the cells, the tumour dies. Moreover, combretastatins disrupt the cell skeleton by causing apoptosis—a programmed massive ‘suicide’ of cells [11,12,13,14,15,16]. At present, combretastatins are being intensively studied as potential drugs, which have led many research groups to seek new synthetic routes for these compounds and their derivatives [11,12,13,14,15,16].

Analysing the structure of Combrestatin, we have found that the core of the combrostatine structure (marked in red in Figure 1) is similar to the structure of di-1,2-diphenylethanol derivatives, which may have different R functional groups. It is known from literature that 1,2-diphenylethanol derivatives are proved to be effective antifungal agents [17,18,19,20,21,22,23,24,25,26,27,28,29,30,31,32,33,34], inhibitors of rhinovirus infection [18,19,20,21,22,23,24,25,26,27,28,29,30,31,32,33,34,35] and potential ligands of oestrogen receptors [19,20,21,22,23,24,25,26,27,28,29,30,31,32,33,34,35,36]. The analysed class of compounds called 1,2-diarylethanes is also a new group of compounds with a similar structure but different functions [17,18,19,20]. These compounds can be tested on bacterial models with different LPS lengths as described in [21,22,23,24].

## 2. Materials and Methods

### 2.1. Microorganisms and Media

Bacterial strains were obtained as a kind gift from Prof. Jolanta Łukasiewicz (Polish Academy of Sciences, Wrocław, Poland). All compounds used in the experiments were the same as described in [18].

### 2.2. Experimental Chemistry

All chemical NMR analyses with the devices for their analysis were identical to those described in [17,18]. Chemical shifts are expressed in parts per million by using TMS as an internal standard.

General Scheme of Reaction Synthesis for Compounds **1**–**18**

Described individual substrates in the synthesis of 1,2-diaryloethanoles are described unchanged as in [25,26,27]. Structural formulas of new petidomimetics are presented in Table 1.

General Procedure for the Synthesis of 1,2-Diarylethyl Esters **4**

To the mixture of arylboronic acid (3 equiv., 0.75 mmol), Pd(0) EnCat^®^ 30NP (5 mol%) and 1,4-benzoquinone (1.2 equiv., 0.3 mmol) in water (1.5 mL) respectively, vinyl ester (1 equiv., 0.25 mmol) was added. The solution was stirred at 40 °C for 48 h, dissolved in methylene chloride (2 × 5 mL) and separated. Combined phase was dried with magnesium sulphate and solvent was evaporated. The products were purified by column chromatography on unmodified silica gel column using hexanes-ethyl acetate gradient. Crude fractions containing the product were combined and evaporated. Structural formulas of the new esters are presented in Table 1.

General Procedure for the Hydrolysis of 1,2-Diarylethyl Ester **4** to Alcohols **7**

A solution of 1,2-diarylethyl ester (1 equiv., 0.2 mmol) and potassium carbonate (2 equiv., 0.4 mmol) in methanol-water mixture (4:1 *v*/*v*, 1 mL) was stirred at room temperature. After 48 h, to the reaction mixture hydrochloric acid was added to reach pH 5. Reaction mixture was extracted with ethyl acetate (3 × 5 mL) and combined organic phase was dried over magnesium sulphate. The solid material was filtered of and organic solvent was evaporated. Crude products were purified on a short silica gel column (hexanes-ethyl acetate mixture), dried under vacuum and characterised. Structural formulas of new alcohols are presented in Table 1.

#### 2.2.1. Product **4a** 1,2-Diphenylethyl Acetate

Yellowish oil. ^1^H NMR (400 MHz, CDCl_3_) δ 7.38–7.15 (m, 8H), 7.15–7.03 (m, 2H), 5.95 (dd, J = 7.9, 6.1 Hz, 1H), 3.20 (dd, J = 13.7, 7.9 Hz, 1H), 3.06 (dd, J = 13.8, 6.1 Hz, 1H), 2.02 (s, 3H); ^13^C NMR (101 MHz, CDCl_3_) δ 170.2, 140.2, 137.2, 129.6, 128.5, 128.3, 128.1, 126.7, 126.7, 76.7, 43.1, 21.3. Elemental analysis: calcd. for C_16_H_16_O_2_: C 79.97%, H 6.71%, found: C 79.76%, H 6.73%.

#### 2.2.2. Product **4b** 1,2-di-p-Tolylethyl Acetate

White powder; mp. 54–55 °C (hexanes:ethyl acetate); ^1^H NMR (500 MHz, CDCl_3_) δ 7.20 (d, J = 8.1 Hz, 2H), 7.14 (d, J = 7.8 Hz, 2H), 7.06 (d, J = 7.9 Hz, 2H), 7.02 (d, J = 8.0 Hz, 2H), 5.91 (dd, J = 8.0, 6.0 Hz, 1H), 3.16 (dd, J = 13.8, 8.0 Hz, 1H), 3.02 (dd, J = 13.8, 6.0 Hz, 1H), 2.33 (d, J = 14.9 Hz, 6H), 2.01 (s, 3H); ^13^C NMR (126 MHz, CDCl_3_) δ 170.3, 137.8, 137.3, 136.1, 134.2, 129.5, 129.1, 129.0, 126.8, 76.8, 42.6, 21.3, 21.3, 21.2. Elemental analysis: calcd. for C_18_H_20_O_2_: C 80.56%, H 7.51%, found: C 80.57%, H 7.74%

#### 2.2.3. Product **4c** 1,2-bis(4-(Trifluoromethyl)phenyl)ethyl Acetate

White solid; mp. 85–86 °C (hexanes:ethyl acetate); ^1^H NMR (400 MHz, CDCl_3_) δ 7.59 (d, J = 8.0 Hz, 2H), 7.52 (d, J = 7.9 Hz, 2H), 7.37 (d, J = 8.0 Hz, 2H), 7.20 (d, J = 7.9 Hz, 2H), 5.98 (dd, J = 7.7, 6.0 Hz, 1H), 3.25 (dd, J = 13.8, 7.7 Hz, 1H), 3.11 (dd, J = 13.8, 6.1 Hz, 1H), 2.05 (s, 3H); ^13^C NMR (101 MHz, CDCl_3_) δ 167.0, 143.6, 140.6, 130.0, 127.0, 125.74, 125.71, 125.67, 125.63, 125.53, 125.50, 125.46, 125.42, 75.6, 42.7, 21.1. Elemental analysis: calcd. for C_18_H_14_F_6_O_2_: C 57.45%, H 3.75% F 30.29, found: C 57.26%, H 3.77%, F 30.15%.

#### 2.2.4. Product **4d** 1,2-bis(4-Chlorophenyl)ethyl Acetate

Grey solid; mp. 74–75 °C (hexanes:ethyl acetate); ^1^H NMR (500 MHz, CDCl_3_) δ 7.30–7.26 (m, 2H), 7.23–7.19 (m, 2H), 7.19–7.13 (m, 2H), 7.02–6.94 (m, 2H), 5.85 (t, J = 6.9 Hz, 1H), 3.14 (dd, J = 13.8, 7.4 Hz, 1H), 2.99 (dd, J = 13.8, 6.5 Hz, 1H), 2.03 (s, 3H); ^13^C NMR (126 MHz, CDCl_3_) δ 170.1, 138.3, 135.1, 134.0, 132.8, 131.0, 128.77, 128.61, 128.13, 75.8, 42.3, 21.2. Elemental analysis: calcd. for C_16_H_14_Cl_2_O_2_: C 62.16%, H 4.56%, Cl 22.93%, found: C 62.10%, H 4.58%, Cl 22.85%.

#### 2.2.5. Product **4e** 1,2-bis(4-Bromophenyl)ethyl Acetate

Semi-solid oil. ^1^H NMR (500 MHz, CDCl_3_) δ 7.47–7.40 (m, 2H), 7.39–7.32 (m, 2H), 7.15–7.07 (m, 2H), 6.98–6.88 (m, 2H), 5.84 (t, J = 6.9 Hz, 1H), 3.12 (dd, J = 13.8, 7.4 Hz, 1H), 2.97 (dd, J = 13.8, 6.4 Hz, 1H), 2.03 (s, 3H); ^13^C NMR (126 MHz, CDCl_3_) δ 170.1, 138.8, 135.6, 131.7, 131.6, 131.4, 128.5, 122.2, 120.9, 75.8, 42.3, 21.2. ESI-MS HR: m/z calcd for C_16_H_14_Br_2_O_2_: 418.9258 [M + Na]^+^, found 418.9262.

#### 2.2.6. Product **4f** 1,2-bis(4-Iodophenyl)ethyl Acetate

Semi-solid oil. ^1^H NMR (500 MHz, CDCl_3_) δ 7.69–7.60 (m, 2H), 7.60–7.51 (m, 2H), 7.03–6.92 (m, 2H), 6.87–6.75 (m, 2H), 5.82 (t, J = 6.9 Hz, 1H), 3.10 (dd, J = 13.8, 7.5 Hz, 1H), 2.95 (dd, J = 13.8, 6.4 Hz, 1H), 2.02 (s, 3H); ^13^C NMR (126 MHz, CDCl_3_) 170.0, 139.5, 137.70, 137.56, 136.3, 131.7, 128.7, 93.9, 92.3, 75.8, 42.3, 21.2. ESI-MS HR: *m/z* calcd for C_16_H_14_I_2_O_2_: 514.8981 [M + Na]^+^, found 514.8964.

#### 2.2.7. Product **4h** 1,2-bis(4-Formylphenyl)ethyl Acetate

Pale yellow solid; m.p. 91–92 °C (hexanes:ethyl acetate); ^1^NMR (400 MHz, CDCl_3_) δ 9.95 (d, J = 8.9 Hz, 2H), 7.85–7.77 (m, 2H), 7.77–7.69 (m, 2H), 7.43–7.34 (m, 2H), 7.26–7.20 (m, 2H), 6.00 (dd, J = 7.6, 6.1 Hz, 1H), 3.27 (dd, J = 13.7, 7.6 Hz, 1H), 3.14 (dd, J = 13.7, 6.1 Hz, 1H), 2.04 (s, 3H).); ^13^C NMR (101 MHz, CDCl_3_) δ 191.8, 191.7, 169.8, 146.1, 143.4, 136.2, 135.3, 130.3, 130.0, 129.8, 127.1, 75.5, 42.9, 21.0. Elemental analysis: calcd. for C_18_H_16_O4: 72.96%, H 5.44%, found: C 72.93%, H 5.42%.

#### 2.2.8. Product **4i** 1,2-bis(4-(Hydroxymethyl)phenyl)ethyl Acetate

White solid; mp. 85–86 °C (hexanes:ethyl acetate); ^1^H NMR (400 MHz, CDCl_3_) δ 7.31–7.23 (m, 4H), 7.21 (d, J = 7.9 Hz, 2H), 7.11–7.03 (m, 2H), 5.92 (dd, J = 7.8, 6.1 Hz, 1H), 4.61 (d, J = 11.1 Hz, 4H), 3.18 (dd, J = 13.8, 7.8 Hz, 1H), 3.04 (dd, J = 13.8, 6.2 Hz, 1H), 2.15 (s, 2H), 1.99 (s, 3H); ^13^C NMR (101 MHz, CDCl_3_) δ 170.3, 140.8, 139.39, 139.31, 136.4, 129.8, 127.09, 126.9, 76.9, 65.1, 65.0, 42.6, 21.3. ESI-MS HR: *m/z* calcd for C_18_H_20_O_4_: 323.1259 [M + Na]^+^, found 323.1259.

#### 2.2.9. Product **4j** 1,2-bis(4-Methoxyphenyl)ethyl Acetate

White solid; mp. 90–91 °C (hexanes:ethyl acetate); ^1^H NMR (400 MHz, CDCl_3_) δ 7.24–7.15 (m, 2H), 7.03–6.95 (m, 2H), 6.88–6.80 (m, 2H), 6.80–6.71 (m, 2H), 5.89–5.79 (m, 1H), 3.78 (d, J = 8.4 Hz, 6H), 3.13 (dd, J = 13.8, 7.6 Hz, 1H), 2.98 (dd, J = 13.8, 6.5 Hz, 1H), 2.01 (s, 3H); ^13^C NMR (101 MHz, CDCl_3_) δ 170.3, 159.4, 158.4, 132.3, 130.6, 129.3, 128.2, 113.84, 113.77, 76.7, 55.37, 55.31, 42.0, 21.4. Elemental analysis: calcd. for C_18_H_20_O4: 71.98%, H 6.71%, found: C 72.01%, H 6.72%.

#### 2.2.10. Product **4k** 1,2-bis(3,4-Dimethoxyphenyl)ethyl Acetate

Brown solid; mp. 101–102 °C (hexanes:ethyl acetate); ^1^H NMR (400 MHz, CDCl_3_) δ 6.87–6.67 (m, 4H), 6.67–6.50 (m, 2H), 5.85 (t, J = 7.0 Hz, 1H), 3.89–3.76 (m, 12H), 3.13 (dd, J = 13.7, 7.3 Hz, 1H), 2.98 (dd, J = 13.7, 6.6 Hz, 1H), 2.03 (s, 3H); ^13^C NMR (101 MHz, CDCl_3_) δ 170.3, 149.0, 148.9, 148.7, 147.9, 132.7, 129.7, 121.8, 119.4, 113.0, 111.2, 111.1, 110.3, 76.8, 56.0, 56.0, 55.9, 42.5, 21.4. ESI-MS HR: *m/z* calcd for C_20_H_24_N_2_O_6_: 360.1573 [M]^+^, found 360.1577.

#### 2.2.11. Product **4l** 1,2-bis(4-((tert-Butoxycarbonyl)amino)phenyl)ethyl Acetate

Brown solid; mp. 155–157 °C (hexanes:ethyl acetate); ^1^H NMR (400 MHz, CDCl_3_) δ 7.29 (d, J = 8.3 Hz, 2H), 7.19 (dd, J = 15.4, 8.3 Hz, 4H), 6.99 (d, J = 8.1 Hz, 2H), 6.51 (d, J = 27.3 Hz, 2H), 5.83 (t, J = 6.8 Hz, 1H), 3.11 (dd, J = 13.7, 7.5 Hz, 1H), 2.96 (dd, J = 13.7, 6.3 Hz, 1H), 1.99 (s, 3H), 1.51 (d, J = 2.5 Hz, 18H); ^13^C NMR (101 MHz, CDCl_3_) δ 170.3, 152.9, 152.8, 138.2, 137.0, 134.6, 131.7, 130.2, 127.6, 118.48, 118.4, 80.7, 80.5, 77.5, 76.5, 42.2, 28.5, 28.46, 21.3. ESI-MS HR: *m/z* calcd for C_26_H3_4_N_2_O_6_: 493.2315 [M + Na]^+^, found 493.2308.

#### 2.2.12. Product **4la** (1E,5E)-1,6-Diphenylhexa-1,5-dien-3-yl acetate

Semi-solid oil; ^1^H NMR (400 MHz, CDCl_3_) δ 7.19–7.38 (m, 10H), 6.64 (d, J = 12.8 Hz, 1H), 6.13–6.22 (m, 2H), 5.52–5.56 (m, 1H), 2.63–2.65 (m, 2H), 2.08 (s, 3H), 1.99 (s, 3H), 1.51 (d, J = 2.5 Hz, 18H); ^13^C NMR (101 MHz, CDCl_3_) δ 170.4, 137.4, 133.2, 132.9, 128.7, 128.1, 127.4, 127.8, 126.3, 80.7, 124.8, 77.5, 74.1, 38.6, 21.5. ESI-MS HR: *m/z* calcd for C_20_H_20_O_2_: 82.16%, H 6.90%, found: C 82.14%, H 6.97%.

#### 2.2.13. Product **4ck** (3,4-Dimethoxyphenyl)-1-(4-(Trifluoromethyl)phenyl)ethyl Acetate

Semi-solid oil; ^1^H NMR (400 MHz, CDCl_3_) δ 7.55 (d, J = 8.4 Hz, 2H), 7.33 (d, J = 15.4, 8.4 Hz, 2H), 6.74 (d, J = 8.0 Hz, 1H), 6.61 (d, J = 8.4 Hz, 1H), 6.47(s, 1H), 5.92 (t, J = 6.8, 1H), 3.85 (s, 3H), 3.75 (s, 3H), 2.95–3.17(m, 2H), 2.079 (s, 3H), 1.51; ^13^C NMR (101 MHz, CDCl_3_) δ 170.1, 148.8, 148.0, 144., 130.4, 13.1, 127.6, 127.1, 125.4, 123.1, 121.8, 112.8, 111.2, 56.0, 55.8, 42.6, 21.3. ESI-MS HR: *m/z* calcd for C_19_H3_19_N_4_F_3_: 391.113 [M + Na]^+^, found 391.1121.

#### 2.2.14. Product **4q** 1,2-Diphenylethyl Benzoate

White solid; mp. 73–73.5 °C (hexanes: ethyl acetate); ^1^H NMR (400 MHz, CDCl_3_) δ 8.12–7.99 (m, 2H), 7.60–7.52 (m, 1H), 7.45 (dd, J = 8.4, 7.0 Hz, 2H), 7.40–7.13 (m, 10H), 6.22 (dd, J = 7.6, 6.0 Hz, 1H), 3.38 (dd, J = 13.8, 7.6 Hz, 1H), 3.23 (dd, J = 13.8, 6.0 Hz, 1H); ^13^C NMR (101 MHz, CDCl_3_) δ 165.8, 140.2, 137.1, 133.1, 130.5, 129.8, 128.52, 128.48, 128.39, 128.1, 126.7, 126.7, 77.4, 43.3.

#### 2.2.15. Product **4t** 1,2-Diphenylethyl 2-Chloroacetate

Pale yellow oil. ^1^H NMR (500 MHz, CDCl_3_) δ 7.35–7.18 (m, 8H), 7.15–7.07 (m, 2H), 6.00 (dd, J = 8.1, 5.9 Hz, 1H), 3.99 (s, 2H), 3.24 (dd, J = 13.9, 8.1 Hz, 1H), 3.10 (dd, J = 13.9, 5.9 Hz, 1H); ^13^C NMR (126 MHz, CDCl_3_) δ 166.5, 139.2, 136.6, 129.7, 128.6, 128.52, 128.5, 126.9, 126.8, 78.8, 42.9, 41.1. ESI-MS HR: *m/z* calcd for C_16_H_15_O_2_Cl 297.0658 [M + Na]^+^, found 298.0654.

#### 2.2.16. Product **4u** (2S)-1,2-Diphenylethyl 2-(6-Methoxynaphthalen-2-yl)propanoate

Brown solid; mp. 96–97 °C (hexanes:ethyl acetate); [α]_D_^20^ = 21.4 (c = 0.55 in CHCl_3_); ^1^H NMR (400 MHz, CDCl_3_) δ 71–60 (m, 2H), 7.55 (dd, J = 30.4, 1.8 Hz, 1H), 7.34–7.09 (m, 9H), 7.09–6.99 (m, 3H), 6.94 (ddd, J = 8.1, 6.3, 1.3 Hz, 1H), 6.87–6.80 (m, 1H), 5.95 (td, J = 8.1, 5.5 Hz, 1H), 3.94 (d, J = 5.3 Hz, 3H), 3.83 (q, J = 7.2 Hz, 1H), 3.17–2.89 (m, 2H), 1.50 (dd, J = 7.2, 6.2 Hz, 3H).; ^13^C NMR (101 MHz, CDCl_3_) δ 170.3, 152.9, 152.8, 138.2, 137.0, 134.6, 131.7, 130.2, 127.6, 118.48, 118.4, 80.7, 80.5, 77.5, 76.5, 42.2, 28.5, 28.46, 21.3. ESI-MS HR: *m/z* calcd for C_28_H_26_O 433.1780 [M + Na]^+^, found 433.1771.

#### 2.2.17. Product **7a** 1,2-Diphenylethanol

^1^H NMR (400 MHz, CDCl_3_) δ 7.44–7.15 (m, 10H), 4.90 (ddd, J = 8.1, 4.9, 3.0 Hz, 1H), 3.05 (dd, J = 13.7, 5.0 Hz, 1H), 3.00 (dd, J = 13.6, 8.4 Hz, 1H), 2.03 (d, J = 2.9 Hz, 1H); ^13^C NMR (126 MHz, CDCl_3_) δ 143.8, 138.1, 129.6, 128.5, 128.4, 127.6, 126.7, 125.9, 75.4, 46.1. Elemental analysis: calcd. for C_14_H_14_O: 81.82%, H 7.13%, found: C 83.87%, H 6.98%.

#### 2.2.18. Product **7k** 1,2-bis(3,4-Dimethoxyphenyl)ethanol

^1^H NMR (400 MHz, CDCl_3_) δ 6.90–6.72 (m, 4H), 6.60 (s, 1H), 6.65 (s, 1H), 4.83–4.80 (m, 1H), 3.87 (s, 6H), 3.85 (s, 3H), 3.82 (s, 3H), 2.9–2.98 (m, 2H); ^13^C NMR (126 MHz, CDCl_3_) δ 149.0, 148.9 148.5, 136.6, 130.6, 118.3, 111.4, 111.0, 109.3, 75.2, 57.3, 57.2, 56.6, 56.0, 45.6. 

### 2.3. Application of MIC and MBC Tests

The minimum inhibitory concentration (MIC) and minimum bactericidal concentration (MBC) analysis are described in detail in [17,18,21,22].

### 2.4. Isolation Plasmids DNA from Bacterial K12, R2–R4 Strains

All four bacterial DNA were isolated according to New England Biolabs, Labjot protocols.

#### 2.4.1. Interaction of the Plasmid DNA from K12 and R4 Strains with 1,2-Diaryloethanols

Analysed diaryloalcohols were described earlier in the literature with coumarin derivatives and α-amidoamides [17,18].

#### 2.4.2. Interaction of the Plasmid DNA from K12 and R4 Strains with Selected Antibiotics

The analysed antibiotics like, kanamycine, streptomycine, ciprofloxacine, bleomycine and cloxacilline have been described in detail in the same way as described earlier in the literature with α-amidoamides [18].

### 2.5. Cleavage of Plasmid DNA by Application with Fpg Glycosylases in Bacterial Cells

The same way as earlier described in the literature [17,18,21,22].

### 2.6. Cleavage of Plasmid DNA by Fpg Protein Modified by Selected Antibiotics

The modified DNA bacterial plasmids modified by selected antibiotics were digested by Fpg enzyme to show similar interaction as the analysed diaryloalcohols. The method by Fpg cleavage is described in detail in literature [17,18].

### 2.7. Statistical Analysis

Parametric analyses using Student’s *t*-test were performed at *p* < 0.05 *, *p* < 0.01 ** and *p* < 0.001 ***.

## 3. Results

### 3.1. Chemistry

New compounds were obtained in synthesis reaction (Figure 2) described in detail in literature [26,27]. The individual 1,2-diaryloethanoles 4 and 7 (shown in Table 1) were isolated via column chromatography.

For the synthesis of target molecules **4,** palladium nanoparticles catalysed reaction was selected [20,21,22,23,24,25,26,27,28,29,30,31,32,33,34,35,36,37], (Figure 2). Preliminary studies on reaction between phenylboronic acid and vinyl acetate showed that commercially available palladium nanoparticles Pd0 EnCat 30NP in the presence of benzoquinone (**3**) proceed in water at room temperature in 1% yield. After optimization, the best condition for reaction was found. Reaction proceeds efficiently in water in the presence of benzoquinone (1.2 equivalent) at 40 °C. Optimal reaction time for reaction requires 48 h. Under optimal conditions compound 4a was obtained in 34% yield (Table 1). Changing arylboronic acids used as a substrate led to the formation of 16 new product **4**, depicted on Figure 3, (associated with specific Log P values) which were obtained in 1 to 34% yield. For selected substrates it was necessary to change catalyst to palladium acetate in order to get respective products (entries in Table 1).

### 3.2. Toxicity of Tested Compounds

MIC and MBC tests are used to analysed 1,2-diaryloethanoles. Both types of MIC and MBC tests are based on the examination of cell membrane damage by the analysed compounds in bacterial cells, which can lead to their death, which is manifested by a colour change of the incubated sample with a positively charged dye solution (resazurin) from dark blue to pink or orange [17,18,21,22]. The dye used penetrates only into dead cells (when the membrane is permanently damaged).

Then the potential between the outside and inside of the membrane is lost. The entire content of the cell is stained, including the cytoplasm and its compartments.

In our research we consider the antibacterial activity of the analysed 1,2-diarylethanols as potential drug candidates, including commonly used antibiotics (Figure 4, Figure 5 and Figure 6). The research was carried out with various reaction products (Table 1), containing in their basic structure two aromatic rings connected with different functional groups in the substituent position, such as e.g., diphenyl, tolylethyl, chlorophenyl, bromo, iodo, formylphenyl, hydroxyphenyl, dimethoxyphenyl, methoxyphenyl.

The antibacterial activity was analysed by MIC and MBC tests [17,18,21,22], Figure 3, Figure 4 and Figure 5 and in the Appendix A in Appendix A.

First, the influence of the groups included as substituents attached to a double aromatic ring (compounds **4a**–**7k**) was investigated. The best MIC and MBC values among all 18 compounds analysed were obtained for nine compounds marked in our study as (**4h**, **4i**, **4k**, **4l**, **4ł**, **4ck**, **4q**, **4u** and **7k**, (average from 0.6 ug/mL for the MIC and 25 µg/mL for MBC in strains R2 and R4). The antibacterial activities of these 1,2-diaryloethanols were determined by the MBC values for the analysed compounds that ranged from 15 to 35 µg/mL (Figure 4). However, the MBC/MIC ratio showed an increase in the value of about 150 to 300 µg/mL in *E.coli* strains R2–R4 compared to the K12 skeleton, where in all annealed tests it was at a very low level, slightly above zero (Figure 4, Figure 5 and Figure 6).

In the first plate (Panel A), on which the K12 strain was applied, the compounds showed a visible colour change at a 10^−7^ dilution, corresponding to a MIC value of 0.0002 mg/mL^−1^ in all reactions analysed. On the second plate (panel B), where the R2 strains were used, colour changes appeared in all analysed compounds (Appendix A in the Appendix A). A visible colour change was already observed at a dilution of 10^−2^ for MIC values of 0.02 mg/mL^−1^ for compounds labelled as **4h**, **4i**, **4k**, **4l**, **41c**, **4ck**, **4q**, **4u** and **7k**. In the case of the remaining analysed compounds, the colour change at the applied dilution of 10^−5^ which corresponds to the MIC value of 0.001 mg/mL^−1^ was very weak (see Appendix A), where the dilution values were similar, but the colour change was more intense than in the R2 skeleton. This proves the destruction of the bacterial membrane and the LPS contained in it. With the length of the LPS, the amount of membrane damage by the compounds may increase. The interactions between the analysed strains and compounds are shown in Figure 3, Figure 4 and Figure 5 and in Appendix A in Appendix A. Increased MIC values were observed for nine compounds **h**, **4i**, **4k**, **4l**, **41**, **4ck**, **4q**, **4u** and **7k** in all analysed R patterns, while very low activity was found for compound K12 for this type of compounds. The highest activity was observed for the strain R4 > R3 > R2 (Figure 4 and Appendix A in the Appendix A). The R4 strain was probably the most sensitive compared to the other strains (Figure 4, Figure 5 and Figure 6). In nine analysed cases, a 48-well plate was used, with the observed MBC/MIC values being approximately 150 times higher than the MIC (Figure 6) and statistically significant as shown in Table 2.

### 3.3. Modification of Plasmid DNA Isolated from *E. coli* R2–R4 Strains with Tested 1,2-Diarylethanoles

Based on the analysis of MIC and MBC toxicity tests on selected bacterial cells from 18 compounds, nine were selected for further analysis, including digestion with Fpg protein on DNA plasmids.

The increase in toxicity in nine analysed compounds (high MIC values) depended on the type of functional groups used in 1,2-diarylethanol (Table 1). We expected a similar effect in plasmid DNA isolated from all model E. coli strains. After treatment with Fpg glycosase, we observed clearly visible damage in the topological changes of the plasmid DNA forms; covalently closed circle (ccc), linear form, open form (oc) and scattered bands (see Figure 7 and Appendix A in the Appendix A). In the modified plasmids, we see significant changes between the control and the modified 1,2-diarylethanes in the electrophoretic images (Appendix A in the Appendix A). This may indicate new substrates for the Fpg protein itself. About 3% or more of the oxidative damage was identified in plasmid DNA after digestion with Fpg protein (Figure 7), which may indicate that the analysed compounds are new substrates for this protein.

In the case of plasmids unmodified with selected compounds from strains K12 and R2–R4, three traditional forms were observed: very weak oc, linear and ccc. 1,2-Diarylethanols, like α-amidoamides and coumarin derivatives, can damage plasmid DNA (strongly changed the topological forms of the plasmids) and the treatment of genetic material induces oxidative stress in living cells. Moreover, the length of the bacterial lipopolysaccharide may influence the microbiological activity of the analysed compounds. The length of the alkyl chain and its structural properties in all analysed diaryloalcohols have a significant impact on their toxicity for the model E. coli R-type strains (after the analysis of the dilution values of 10^−2^ or 10^−3^ mL and the corresponding concentrations of 0.02 or 0.002 mg/mL^−1^ in all analysed strains in the MIC and MBC tests). The analysed relationships were statistically significant with *p* < 0.05 *.

The results of bacterial DNA modified with 1,2-diarylethanes are presented in Appendix A; seven showed all analysed compounds with different substituents containing: 1,2-bis(4-formylphenyl)ethyl acetate, 1,2-bis(4-(hydroxymethyl)phenyl)ethyl acetate, 1,2-bis(3,4-dimethoxyphenyl)ethyl acetate, 1,2-bis(4-((tert-butoxycarbonyl)amino)phenyl)ethyl acetate, (1E,5E)-1,6-diphenylhexa-1,5-dien-3-yl acetate, -(3,4-dimethoxyphenyl)-1-(4-(trifluoromethyl)phenyl)ethyl acetate, 1,2-diphenylethyl benzoate, (2S)-1,2-diphenylethyl 2-(6-methoxynaphthalen-2-yl)propanoate, 1,2-bis(3,4-dimethoxyphenyl)ethanol.

Therefore, the next step in our research was the use of 1,2-diarylethanols as potential analogues of commonly used antibiotics. The experimental set-up was similar with the 1,2-diarylethanol used by the MIC and MBC assays, both in quantity and concentration (Figure 8), and with plasmids isolated from bacteria and modified with appropriate antibiotics; kanamycin, streptomycin, ciprofloxacin, bleomycin and cloxacillin and digested with Fpg protein. In all analysed R-type strains, using the MIC and MBC tests, a colour change was observed for all the antibiotics used at a dilution of 10^−3^ (Figure 8, Appendix A in the Appendix A), which corresponds to 0.02 mg mL^−1^ in the analysed MIC (Appendix A, Appendix A). The effect of the analysed MIC test for antibiotics was observed in all tested strains, but the greatest effect was observed for cloxacillin in the R4 strain, where the colour change on the plate was already at a dilution of 10^−1^, which corresponds to the value of 0.2 mg/mL^−1^. Also high values were observed after bleomycin treatment in all model strains where the colour change on the plate was already visible at a dilution of 10^−2^, which corresponds to a value of 0.02 mg/mL^−1^. High MIC values were observed for ciprofloxacin, streptomycin and kanamycin in strains R3 and R4 (Figure 8). It has also been shown that the R4 strain, having the longest LPS, interacts with all the active groups contained in antibiotics. The obtained results were statistically significant at *p* < 0.05.

Based on the action of 1,2-diarylethanols and the selection of nine compounds for further studies with the Fpg protein, a similar experimental system was used for the analysed antibiotics in terms of dose and concentration (Figure 9, Appendix A, see Appendix A). The bacterial DNA isolated from all model strains and ‘reacted’ with a panel of five antibiotics after digestion with Fpg protein, the cc, linear and oc forms were observed in different proportions (Appendix A
Appendix A).

Additionally, in bacterial DNA isolated from the R2 strain after treatment with antibiotics and additionally with the Fpg protein after the cloxacillin treatment, two additional forms appeared above the ccc form, which proves the formation of concatamers (additional looped structures in DNA). Streaks appeared in the samples digested with the protein alone as a result of digestion with the protein recognizing oxidative damage on the plasmid, seen as ‘smir’. Plasmid DNA isolated from the R3 strain was slightly disturbed because of the modification with antibiotics, the samples that were not digested with Fpg protein, referred to as controls, migrated as one common sample, difficult to separate on agarose gel. After digestion with Fpg protein in all analysed cases after the action of antibiotics and especially after the action of bleomycin and cloxacillin, the ratio of topological forms of the analysed plasmids was disturbed and was visible as a very weak form of ccc together with a barely visible linear form, while the form of oc migrated significantly below the value of both the previous forms and was as a blurry band. A similar but stronger effect was seen on a plasmid isolated from the R4 strain where the effect of migration of forms in the form of blurred bands after the treatment with the Fpg protein was even more visible. This proves the very strong and strong effect of these antibiotics on the genetic material itself contained in the cell and the destruction of the bacterial membrane in the form of a colour change in the analysed MIC and MBC tests (Appendix A in Appendix A).

The highest values of observed damage in bacterial DNA were observed for the antibiotics cloxacillin > bleomycin > ciprofloxacin > streptomycin > kanamycin. The sensitivity of the analysed strains and bacterial DNA isolated from them to the analysed antibiotics was as follows: R4 > R2 > R3 > K12. Values of oxidative damage in plasmid DNA after digestion with Fpg protein and treatment with cloxacillin and bleomycin were five times higher in comparison to other antibiotics in all analysed strains and ranged from 2.5 to 3.5 percent. These values were very close to the Fpg protein digestion of the selected 1,2-diarylethanols, where the level of damage was similar. Damage values in plasmid DNA after digestion with Fpg protein for ciprofloxacin, kanamycin and streptomycin were similar for the model strains K12, R2 and R3 and were 0.5%. Digestion of the modified bacterial DNA with the Fpg protein increased the intensity of the bands in all isolates from the analysed strains. This shows that the microbial toxicity of the targeted antibiotics to 1,2-diarylethanols is significant and affects the structure and length of LPS in bacteria. Also, the endogenous level of Fpg glycosylase in unmodified bacteria is very low, so it is possible that all guanine residues have not been fully repaired in the plasmid DNA and may compete with one or more repair enzymes in the base excision system (BER). Bacteria recruit specific topoisomerases that allow for structural relaxation and access to modified DNA bases. This suggests that stabilization of the topoisomerase cleavage complex is essential for the cell as it blocks replication and transcription. In addition, the secondary stability effect of 8oxoG and its derivatives, as well as other modifications to oxidized bases such as Fapy Ade or Fapy Gua in the genome, may affect the global amount of super-helical DNA.

## 4. Discussion

Determination of the antibacterial action of compounds that can potentially be used in vitro as chemotherapeutic agents for bacterial cells is important to establish the proper action and safety of the drug. The use of bacterial cells of selected model strains containing LPS of different lengths in their structure in cytotoxicological studies has many positive advantages [17,18,21,22]. These include the short duration of experiments, simple methodology used in the research, greater precision in the study of cellular processes with the use of molecular biology and microbiological engineering methods, repeatability of research experiments and interpolation of research results on human cells.

In our research, we observed that out of all 18 compounds used on bacterial cells, only nine were toxic. The antibacterial efficacy of the selected compounds used was very high after the applied tests. The highest values in both types of tests were observed for the compounds marked as **4h**, **4i**, **4k**, **4l**, **4k**, **4ck**, **4q**, **4u**, **7k**. Conversely, it reduces compounds such as: **4a**, **4b**, **4c**, **4d**, **4e**, **4f**, **4j**, **4t**, **7a**. This demonstrates that the type of the substituent and the two aromatic rings are linked together. In the analysed 1,2-diarylethanols which contain different types of substituents in their structure, a significant increase in cytotoxicity may occur in relation to the subsequent analysed model strains of *E. coli* R2 > R3 > R4 differing in the length of the LPS. It is known from the literature that the analysed strains can cause diseases related to circulatory and digestive system disorders, often leading to the development of various types of cancer [23,24].

The potential toxicity of those selected from all tested 1,2-diarylethanols for all analysed bacterial cells was high, and the highest values were noted for strains R2 and R4 against strains R3 and K12 [17,18,21,22,24], for compounds **4h**–**7k** containing functional groups consisting of 1,2-bis(4-formylphenyl)ethyl acetate, 1,2-bis(4-(hydroxymethyl)phenyl)ethyl acetate,1,2-bis(3,4-dimethoxyphenyl)ethyl acetate, 1,2-bis(4-((tert-butoxycarbonyl)amino)phenyl)ethyl acetate,(1E,5E)-1,6-diphenylhexa-1,5-dien-3-yl.acetate,(3,4-dimethoxyphenyl)-1-(4-(trifluoromethyl)phenyl)ethyl acetate, 1,2-diphenylethyl benzoate, (2S)-1,2-diphenylethyl 2-(6-methoxynaphthalen-2-yl)propanoate, 1,2-bis(3,4-dimethoxyphenyl)ethanol (Table 1, Figure 4 and Figure 5) for the analysed strains [17,18,21,22,24]. 1,2-diarylethanols with different functional groups were more effective in R than K12. The effect of the interaction of the analysed compounds with the bacterial membrane was very similar to the interaction of ionic liquids containing quaternary ammonium surfactants [17,18,38,39,40].

Changes in the activity of bacterial cells due to the given compounds probably result from the rearrangement of the polarity of the components of the bacterial membrane containing LPS as a result of interaction with 1,2-diarylethanols exhibiting strong cytotoxicity, inducing oxidative stress in the bacterial cell, leading to its damage and biochemical decomposition of genetic material [21,22,24,38,39,40].

The obtained results constitute the basis for the continuation of further research on other pathogenic bacterial strains associated with diseases of the vital systems related to the basic functioning of the human body. They will also allow to identify potential mechanisms of degradation in cell membranes through new synthesized substances as new precursors of known and commonly used antibiotics [21,22].

The analysis of the toxicity of 1,2-diarylethanols used in our research shows that it is strongly related to the length of the LPS in the analysed types of bacteria R2–R4. In addition to the MIC and MBC tests, the digestion analysis of the modified plasmids isolated from strains K12 and R2, R3 and R4 by FPg protein with N-glycosylase/AP lyase activity was performed [17,18,21,22,24].

Percentage of damage to plasmids modified by 1,2-diarylethanols was determined from the changes in the topological forms of ccc, linear and oc after treatment with the Fpg protein. From literature data Fpg protein is known to have a wide range for recognising and eliminating oxidised and alkylated bases that have been modified by ROS or RNS. Fpg protein is now recognised as sensitive by many international laboratories ‘marker’ of oxidized bases formed in bacterial cells under the influence of oxidative stress caused by internal and external factors. It is assumed that the amount of identified base damage above 3–4% in a single or double strand of DNA caused by oxidation or alkylation by the enzyme Fpg is a very important indicator of the degree and strength of guanine or adenine modification in the analysed genetic material [17,18,28].

In our research, protein-Fpg recognises modifications introduced by 1,2-diarylethanols to plasmid DNA, especially for nine compounds selected from all tested compounds. Visible changes between topological forms and the emerging so-called ‘Smear’ of bands after digestion of modified plasmids with protein are a result of providing the enzyme with new potential substrates for its repair activity as glycosylases [28].

The results suggest that the selected compounds modify the bacterial DNA isolated from the K12 and R model strains and are recognised by Fpg glycosylase (Appendix A). The most effective results are given by the compounds **4h**, **4i**, **4k**, **4l**, **4ł**, **4ck**, **4q**, **4u**, **7k**. This means that in the future, specific 1,2-diarylethanols may be designed as new potential substitutes for antibiotics with very similar chemical structure, but stronger activity against all bacterial strains. According to literature data, frequent chemotherapy with antibiotics such as ciprofloxacin, kanamycin, streptomycin and bleomycin resulted in immunisation of many bacterial pathogens [29]. Therefore, we additionally tested cloxacillin, a component of the drug syntarpen, which is administered to patients in the case of acute gastrointestinal or cardiovascular inflammation by bacteria of the genus Stahylococcus areus [30]. This antibiotic has a very broad spectrum of activity against Gram-negative bacteria, including E. coli [31], as shown in the 48-well MIC plates (Appendix A in the Appendix A). After treatment of bacterial DNA with a set of five antibiotics and additionally digested with Fpg protein, the highest level of damage was observed for the strains R4 > R2 > R3 > K12. These values were two times higher after treatment with bleomycin and cloxacillin compared to other antibiotics in all analysed strains (Appendix A in the Appendix A). The toxicity values of the analysed ciprofloxacin, kanamycin and streptomycin were similar, but only for the K12, R2 and R3 strains. In the case of the R4 strain, a 2.5-fold increase in value was observed after treatment with kanamycin compared to the other antibiotics, ciprofloxacin and streptomycin. Therefore, it is reasonable to look for compounds that have a similar structure but stronger biological and chemical properties that would be more toxic to bacterial cells.

Based on the analysis of MIC and MBC tests, we can design 1,2-diarylethanols with very high toxicity to Gram-negative bacterial cells, similar to the previously analysed α amidoamides and coumarin derivatives [17,18]. The practical application of the analysed compounds will allow their use in the future as potential new ‘antibiotics’, which will be more toxic and effective against all strains of pathogenic bacteria.

A multi-component reaction is used to discover new properties of the selected nine compounds as potential drugs, presented in Table 1. The studies showed that the Fpg digested samples showed a ‘large smear’ and rearrangement of the topological forms of the plasmid on plates R3 and R4 (Appendix A in the Appendix A). This means that enzymes recognised damage caused by modification by antibiotics that are ‘toxic’ to DNA in the same way as for bacterial DNA modified with selected 1,2-diarylethanes.

The structure of Fpg, which determines its dual activity as protein glycosylase and AP lyase, consists of two domains. N-terminal domain containing the active site within the first 72 amino acid residues and the secondary amino group containing Pro1 and Glu2 as well as Lys56 and Lys154. Pro1 and Glu2 are essential for the activity of the Fpg enzyme to recognise the modified bases. The C-terminal domain contains a helix-harpin-helix (HhH) motif and is involved in DNA binding. Functional homologues of formamidopyrimidine DNA glycosylase have now been identified in Saccharomyces cerevisiae (yOGG1) and humans (hOGG1) [32,33].

In bacterial DNA isolated from the R4 strain digested with the Fpg protein modified with diaryl alcohol and after modification with antibiotics (including cloxacillin), we observed the disappearance of CCC forms and the appearance of a single strand migrating similarly, but slightly slower than the OC form in the modified bacterial DNA, creating a complex with a large molecular weight (see Appendix A). The effect of band migration in both experiments was similar to each other, which further strengthens the argument that selected 1,2-diarylethanols can replace antibiotics in the future, enhancing their biological functions by affecting the bacterial cell membrane.

This is indicated by the results of digestion with the Fpg protein (Appendix A), which forms a strong complex in the form of concatamers with modified DNA plasmids. This complex is strong enough and stable in electrophoresis. He suggests that the Fpg protein was covalently linked to DNA via amino acids or high molecular weight protein fragments. This suggests that the Fpg protein recognizes some induced DNA damage following modification by 1,2-diarylethanols and antibiotics, including cloxacillin.

This suggests that protein–DNA cross-linking by aldehyde groups of the resulting adducts may be the proper mechanism for their formation, but the extent of this may vary from protein to protein. The repair of the analysed three topological forms of plasmids in DNA in the case of the compounds used is carried out by the BER system, which includes Fpg glycosylase. Then, the aforementioned DNA–protein complexes are formed, visible as blurred bands or compact forms in the form of concatamers, which play slower during agrose gel electrophoresis.

We postulate that the interaction between DNA and 1,2-diarylethanes and antibiotics occurs through a covalent bond between aldehyde groups and other groups present as substituents on the aromatic double rings that can arise in single-stranded plasmid DNA. Each band visible on the gel represents the nucleotide positions at which strand damage or breakage was induced by digestion with DNA glycosylase/AP lyase. Analysis of the mapped sites recognized by the Fpg enzyme suggests that all DNA residues are linked to them as modified oxidized bases that may correspond to the 8-oxoguanine centres in the matrix. Probably the spatial rearrangement of the bases and the selection of the appropriate modified base by the analysed compounds are as a result of the rearrangement of the chain, which is effectively digested by the Fpg glycosase, which is a new substrate for them. The addition of specific 1,2-diarylethanol groups causes thermodynamic destabilisation of a single helix on cytosine and guanine [32,33] by loosening the compact structure of CG-rich regions that were more easily bypassed by Fpg enzymes, allowing the identification of damaged cytosine residues. The nature of the DNA damage mechanism recognised by Fpg glycosylase requires further clarification. In the Fpg protein, two lysine residues, Lys-57 and Lys-155, take part in catalysis and directly interact with 8-oxopurines [33]. Moreover, the Fpg Pro2 protein is involved in the breaking of the N-glycosidic bond by forming a Schiff base from C1 deoxyribose. This suggests that the N-terminal amino group is most likely available for binding specific groups of components and antibiotics [32,33]. In general, the R4 strains showed a higher sensitivity than the R2 and R3 strains (Figure 4, Figure 5, Figure 6, Figure 7, Figure 8 and Figure 9, Appendix A). Our research shows that both the MIC and MBC tests used were successfully applied in all analysed compounds (Figure 4, Figure 5 and Figure 6). The analysed LPS-containing bacterial strains showed increased sensitivity to the toxic effects of 1,2-diarylethanols with varying severity (as compared to the control strain K12). The R4 strain, due to the longest LPS length, was probably the most sensitive. Studies of cytotoxic activity on bacterial cells of the newly synthesized compounds will allow for a better selection of the type and type of substituents, leading to the creation of drugs with the best biological parameters and microbiological activity for the analysed bacterial cells. The newly obtained compounds are a new alternative as potential innovative substitutes for antibiotics due to their specific structure related to commonly used drugs.

## 5. Conclusions

The conducted study confirms the usefulness of mass screening tests for inhibitors of specific bacteria, such as 1,2-diarylethanols that induce plasmid DNA damage, in order to identify the antibiotics with a new and innovative way of action—acting on all types of bacteria.

In our experiments, we found that:1,2-diarylethanols with a specific structure are able to modify all strains of *E. coli* (R2–R4) and their plasmid DNA, then the spatial structure of LPS contained in their cell membrane is changed.The R4 type strain was the most sensitive amongst all the tested *E. coli* strains.The interaction of the analysed 1,2-diarylethanols with the cell membrane of the K12 strain shows differences in the O-antigen and the shortened oligosaccharide core compared to the analysed R-type strains, which may play an important role in the cellular response to the charged compounds.The toxicity of aromatic groups together with the alkyl substituents depends on their interaction with the membrane that may become involved in the structures of cell walls and change their hydrophobicity.Changes in the structure of the bacterial membrane and disturbances in its integrity may result in changes in the bacterial response to other biologically active compounds such as antibiotics.Plasmid DNA damage has been linked to the structure of verified compounds, which suggests that the presence of 1,2-diarylethanoles affects the bacteria LPS and generates oxidative stress, as we have already observed in our previous studies [17,18].The studied 1,2-diarylethanols show a different effect in the MIC and MBC tests, which is strongly correlated with the steric factor of functional groups in the form of substituents with a short chain alkyl in the structure of analysed compounds [17,18].

The results of our new experiments are important to understanding biological properties of the tested -1,2-diarylethanols as a function of new potential antibiotics and their toxic effect on Gram-negative bacteria cells in the face of growing resistance to various drugs and epidemic infections.

## Figures and Tables

**Figure 1 materials-14-01025-f001:**
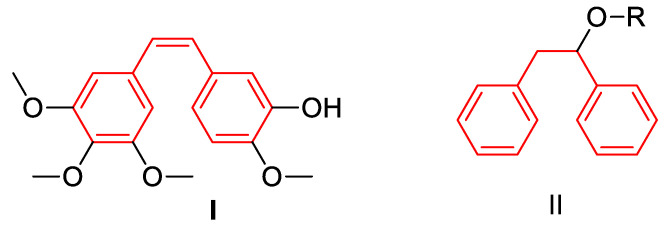
Structure of combrestatine **I** and proposed analogues **II.**

**Figure 2 materials-14-01025-f002:**
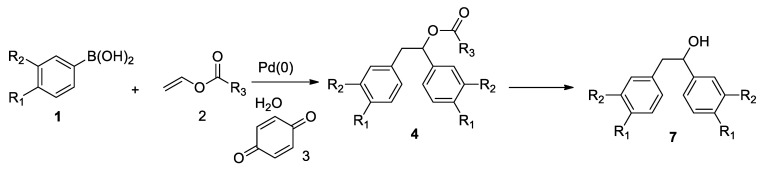
Scheme of palladium nanoparticles catalysed reaction leading to esters **4** in the presence of benzoquinone (**3**) followed by hydrolysis to alcohols **7**.

**Figure 3 materials-14-01025-f003:**
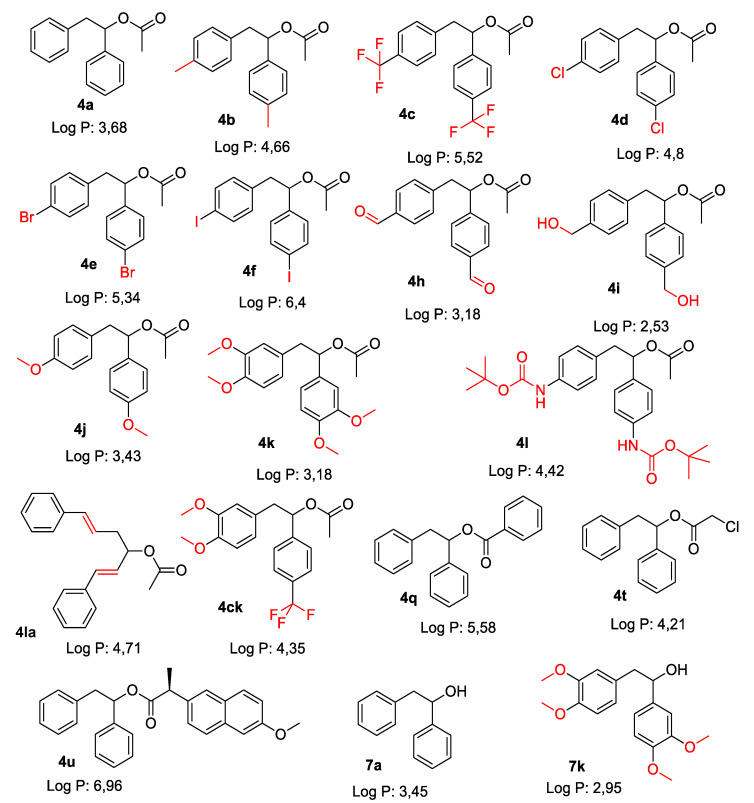
The chemical structure of 1,2-diarylethanols obtained. The specific substituents that determine the reactivity of the compounds are marked in red.

**Figure 4 materials-14-01025-f004:**
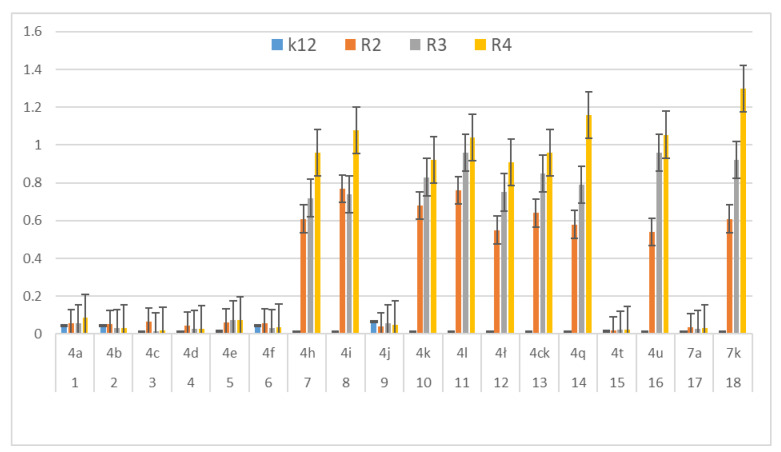
Minimum inhibitory concentration (MIC) of the diaryloalcohols in model bacterial strains. The x-axis compounds **4a**–**7k** were used sequentially. The y-axis shows the MIC value in mg mL^−1^.

**Figure 5 materials-14-01025-f005:**
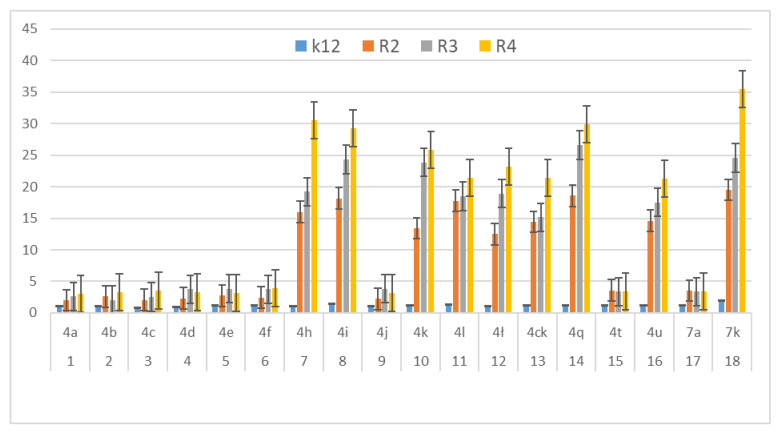
MBC of the diaryloalcohols in model bacterial strains. The x-axis compounds **4a**–**7k** were used sequentially. The y-axis shows the MBC value in mg mL^−1^.

**Figure 6 materials-14-01025-f006:**
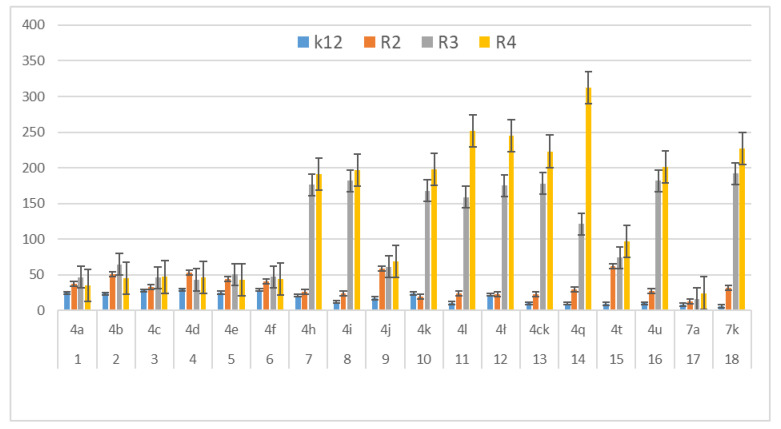
MBC/MIC of the diaryloalcohols in model bacterial strains. The x-axis compounds **4a**–**7k** were used sequentially. The y-axis shows the MBC/MIC value in mg/mL^−1^

**Figure 7 materials-14-01025-f007:**
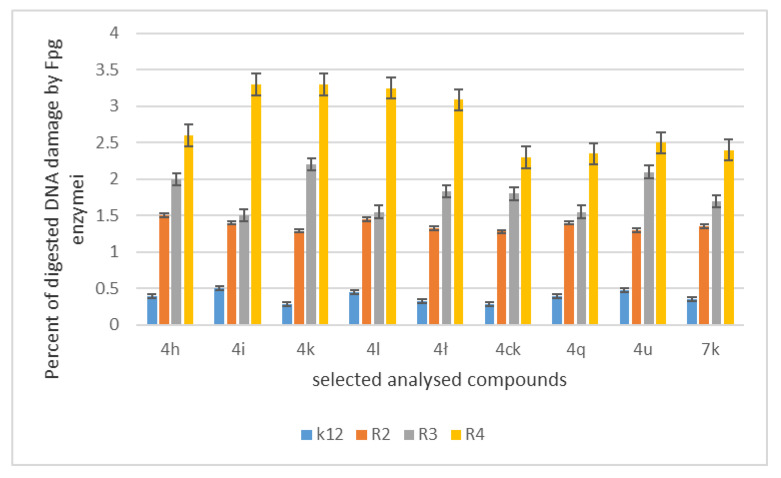
Percent of digested DNA damages recognised by Fpg enzyme- (y-axis) with control K12 and. R2–R4 strains (x-axis); The selected compounds **4h**–**7k** were statistically significant at *p* < 0.05.

**Figure 8 materials-14-01025-f008:**
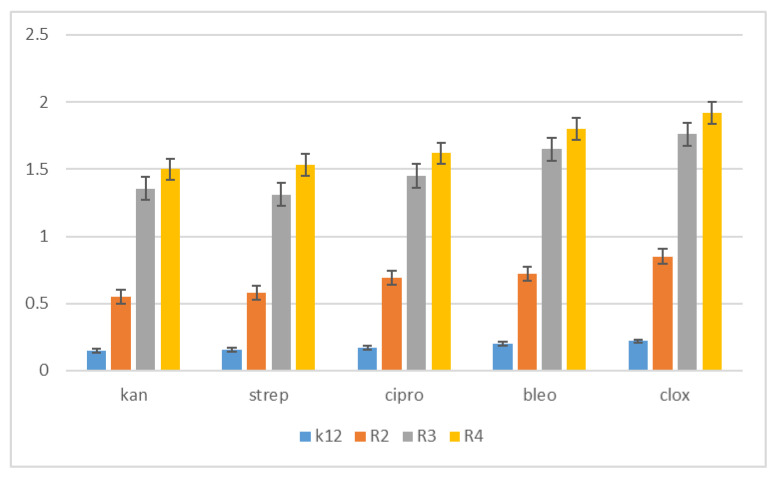
Examples of MIC with model bacterial strains on studied antibiotics; kanamycine, streptomycine, ciprofloxacine, bleomycine and cloxacilline. The x-axis features antibiotics used sequentially. The y-axis features the MIC value in mM.

**Figure 9 materials-14-01025-f009:**
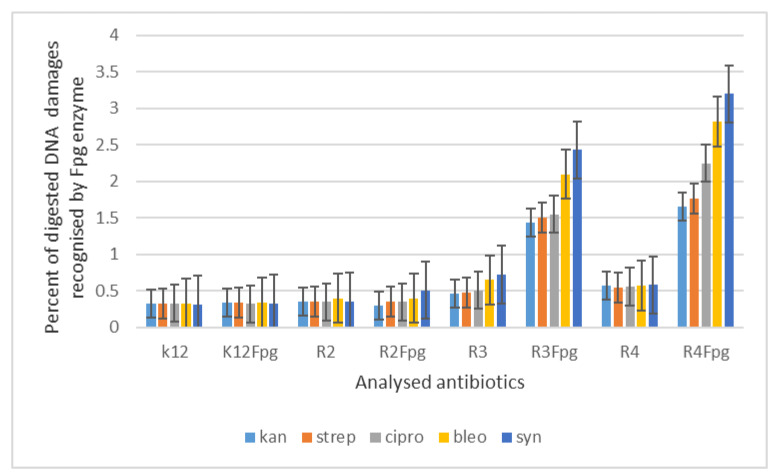
Percentage of bacterial DNA recognized by Fpg enzyme in model bacterial strains after kanamycine, streptomycine, ciprofloxacine, bleomycine and cloxsacilline treatment. The compounds were statistically significant at *p* < 0.05.

**Table 1 materials-14-01025-t001:** ^1^ Yield of reaction calculated by ^1^H NMR mixtures. ^2^ Reaction catalysed by Palladium acetate. Base catalysed hydrolysis of ester **4a** and **4k** led to the formation of alcohols **7a** and **7k** in over 75% yield.

Compound Symbol	R1	R2	R3	Yield ^1^ (%)
4a	H	H	Me	34
4b	Me	H	Me	19
4c	CF_3_	H	Me	2
4d	Cl	H	Me	34
4e	Br	H	Me	6
4f	I	-	Me	**0(2^2^)**
4h	-CHO	H	Me	4
4i	-CH_2_OH	H	Me	5
4j	CH_3_O-	H	Me	1
4k	CH_3_O-	CH_3_O-	Me	33
4l	Me3COC(O)NH-	H	Me	9
4la	-	H	Me	**0(6^2^)**
4ck	CF_3-,_ CH_3_O-	CF_3-,_ CH_3_O-	Me	2
4q	H	H	Ph	22
4t	H	H	Cl	7
4u	H	H	naproxen	**0(21^2^)**

**Table 2 materials-14-01025-t002:** Statistical analysis of all analyzed 18 compounds at *p* < 0.05 *, < 0.01 **, < 0.001 *** in MIC, MBC and MBC/MIC tests. Diaryloalcohols **4a**–**7k** were used sequentially.

No of Samples	4a	b	c	d	e	f	h	i	j	k	l	ł	ck	q	t	u	7a	7k	Type of Test
K12							***	***		**	**	***	**	***		***		**	MIC
R2							***	***		**	**	***	**	***		***		**	MIC
R3							***	***		**	**	***	**	***		***		**	MIC
R4							***	***		**	**	***	**	***		***		**	MIC
K12							**	**		***	***	**	***	**		***		***	MBC
R2							**	**		***	***	**	***	**		***		***	MBC
R3							**	**		***	***	**	***	**		***		***	MBC
R4							**	**		***	***	**	***	**		***		***	MBC
K12							*	*		**	**	*	**	*		***		*	MBC/MIC
R2							*	*		**	**	*	**	*		***		*	MBC/MIC
R3							*	*		**	**	*	**	*		***		*	MBC/MIC
R4							*	*		**	**	*	**	*		***		*	MBC/MIC

## Data Availability

On request of those interested. The manuscript also contains one quotation from the JAFS journal.

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
