# Peer review of "1,2-Diarylethanols—A New Class of Compounds That Are Toxic to E. coli K12, R2–R4 Strains"

_materials, 2021, doi:10.3390/ma14041025_

Round 1
Reviewer 1 Report
Authors describe the potential of 1,2-diaryl ethanols as compounds toxic to E. coli strains which differ in length of LPS. MIC and MBC tests, as well as tests for DNA damage, were used for the biological evaluation. The same methodology authors have used in previous papers. Research is interesting and used methods are appropriate. However, the overall impression is that the authors did not pay enough attention in the preparation of the manuscript. The main drawback of the work is an insufficient explanation of the structural motif used in the design of compounds toxic to E. coli. In a previous article (Materials 2020), the authors described α-aminoamides as peptidomimetics. However, prepared 1,2-diaryl ethanols do not have a similar structure, and therefore it should explain in the text why prepared compounds are peptidomimetics. Furthermore, the authors were inspired by a natural compound combrestatine with pronounced anti-cancer activity, while they are focused on bactericidal activity of compounds.
The article contains a number of spelling mistakes, including unfinished sentences, and inconsistent usage of terms. The following should be also corrected:
- E. coli written in italic,
- IUPAC nomenclature should be used (for example - N,N italic in N,N-dimethylformamide, tert-, p-, E-, symbol α instead alpha…)
- measurement units need to be checked in the text and in figures (for example mL and ml are used, then ml-1 or / ml and etc…), dilution, p should be in italic, etc.)
Authors should correct manuscript thoroughly.
Other comments:
In Abstract authors should pay attention to “-Fpg glycosylases.” and inconsistency in writing 1,2- diaryl ethanols (assimilate with the title).
Mass spectrometry is mentioned in 2.2. Experimental chemistry, while MS data is not given for the characterization of compounds. For new compounds MS date is mandatory. The ppm units are missing in NMR data.
Figure 2 lacks information regarding R1, R2, and R3 substituents (R1, R2, and R3 should be added in Figure 2).
The description of Table 1 title should be corrected, and column “Entry” in the table is redundant. Footers 1 and 2 should be listed under Table and not in the title. The use of footer in the table should be adjusted since entries 6 and 12 are more difficult to understand.
In Figure 3 logP values are shown and they are not commented in the text.
Graphic representations of MIC and MBC tests lack y-axis marks. In figures 4, 5, and 6, the units of measurement should be checked (Figure 6 shows MBC / MIS ratio and therefore there are not units).
In Table 2, the confusing way of presenting date; numerical listing of compounds is not consistent with the numbers of compounds used in the experimental part, this needs to be harmonized.
The conclusion should be summarized and written more concisely.
Author Response
RESPOND TO REVIEWERS COMMENTS:
Thank you very much for valuable suggestions that have contributed to the improvement of the quality of work.
Reviewer 1
Authors describe the potential of 1,2-diaryl ethanols as compounds toxic to E. coli strains which differ in length of LPS. MIC and MBC tests, as well as tests for DNA damage, were used for the biological evaluation. The same methodology authors have used in previous papers. Research is interesting and used methods are appropriate. However, the overall impression is that the authors did not pay enough attention in the preparation of the manuscript.
The main drawback of the work is an insufficient explanation of the structural motif used in the design of compounds toxic to E. coli. In a previous article (Materials 2020), the authors described α-aminoamides as peptidomimetics.
However, prepared 1,2-diaryl ethanols do not have a similar structure, and therefore it should explain in the text why prepared compounds are peptidomimetics.
We agree with the suggestion, so we changed the word peptidomimetics in the title to the word compounds. We therefore consider it more legible and concise for the recipient. We also removed all words related to peptidomimetics from the manuscript.
Furthermore, the authors were inspired by a natural compound combrestatine with pronounced anti-cancer activity, while they are focused on bactericidal activity of compounds.
The biological activity of combrestatine was recently reviewed in respect to antioxidant, anti-inflammatory and antimicrobial effects.. (Molecules 2020, 25(11), 2560; https://doi.org/10.3390/molecules25112560). Antimicrobial and Leishmanicidal Activities were presented and discussed. This information was missed and now is included in manuscript body.
The article contains a number of spelling mistakes, including unfinished sentences, and inconsistent usage of terms. The following should be also corrected:
- E. coli written in italic,
The names of the bacteria have been corrected in italics
- IUPAC nomenclature should be used (for example - N,N italic in N,N-dimethylformamide, tert-, p-, E-, symbol α instead alpha…)
The IUPAC nomenclature has been used
- measurement units need to be checked in the text and in figures (for example mL and ml are used, then ml-1 or / ml and etc…), dilution, p should be in italic, etc.)
The measurements units need to be checked in the text and figures
Authors should correct manuscript thoroughly.
Other comments:
In Abstract authors should pay attention to “-Fpg glycosylases.” and inconsistency in writing 1,2- diaryl ethanols (assimilate with the title).
Comments have been included in the abstract
Mass spectrometry is mentioned in 2.2. Experimental chemistry, while MS data is not given for the characterization of compounds. For new compounds MS date is mandatory. The ppm units are missing in NMR data.
Chemical part was modified by reviewer suggestion. The statement on ppm scale in NMR experiment was introduced into Part 2.2 Experimental Chemistry. For all new compounds elemental analysis or high resolution mass spectra are provided.
Figure 2 lacks information regarding R1, R2, and R3 substituents (R1, R2, and R3 should be added in Figure 2).
The information regarding R1, R2, and R3 substituents are presented in Table 1. We assumed that in the presented form (figure and table 1 complement each other) and they will be more legible
The description of Table 1 title should be corrected, and column “Entry” in the table is redundant. Footers 1 and 2 should be listed under Table and not in the title. The use of footer in the table should be adjusted since entries 6 and 12 are more difficult to understand.
Table 1 has been corrected
In Figure 3 logP values are shown and they are not commented in the text.
has been included in the text
Graphic representations of MIC and MBC tests lack y-axis marks. In figures 4, 5, and 6, the units of measurement should be checked (Figure 6 shows MBC / MIS ratio and therefore there are not units).
The descriptions of the x and y axes are under the graphs, we thought that would make them more readable. The MBC / MIC ratio follows the value given in the graphs.
In Table 2, the confusing way of presenting date; numerical listing of compounds is not consistent with the numbers of compounds used in the experimental part, this needs to be harmonized.
Table 2 has been corrected (numerical listing of compounds are now consistent with the numbers of compounds used in the experimental part, now is harmonized.
The conclusion should be summarized and written more concisely.
The conclusion has been corrected

Reviewer 2 Report
Dear authors,
The authors made a significant improvement in the Manuscript. I suggest to accept this Manuscript (ID : materials-1110756) in present form.
English language and style are fine/minor spell check required. For example, in Abstract:
- A) my suggestion
“….after application of a repair enzyme (Fpg glycosylases) was analyzed”.
- B) instead of
“…after application of a repair enzyme was analyzed. - Fpg glycosylases” .

Author Response
RESPOND TO REVIEWERS COMMENTS:
Thank you very much for valuable suggestions that have contributed to the improvement of the quality of work.
Reviewer 2
Open Review
Dear authors,
The authors made a significant improvement in the Manuscript. I suggest to accept this Manuscript (ID : materials-1110756) in present form.
English language and style are fine/minor spell check required. For example, in Abstract:
- A) my suggestion
“….after application of a repair enzyme (Fpg glycosylases) was analyzed”.
- B) instead of
“…after application of a repair enzyme was analyzed. - Fpg glycosylases” .
The abstract has been corrected as suggested. The amendments are marked in green

Reviewer 3 Report
The authors presented interesting work in which they described the development of novel antimicrobial compounds based on the 1,2-diarylethanol scaffold.
The article is well designed and all experiments are conducted according to the good scientific level.
However, I have several issues that should be solved before the article will be accepted for the publication.
- I think that use of the word: “peptidomimetic” is misleading. Peptidomimetic is a compound which mimics the peptide structure for the better pharmacokinetic behavior of the molecule. I did not see in the paper any peptide-like structure as a precursor for the design of 1,2-diarylethanols.
- The authors wrote that: “ In this study, we paid attention to the determination of the biological activity of the analyzed 1,2-diaryloethanoles as new drug substitutes…”. I think that better is to use the word: drug candidates and not “new drug substitutes”.
- If authors are describing in the introduction the use of alcohols as an antiseptic agents like the basis for the development of the 1,2-diarylethanols, they should be consistence with that and they should tell that also their molecules will be used as a sprayers for the cleaning of the surfaces and hands but not as antibiotics. In this case, use of the classical antibiotics as kanamicyn for the positive control is not appropriate.
- I did not see in the Supplemental Information all spectra images of synthesized compounds, as it is acceptable in the chemistry journals.
- English should be improved also, by editing the paper by professional scientific English editor.
- Also the article has a lot of typos, for example:
- Changes in the structure of the bacterial membrane and disturbances in its integrity may result in changes in the bacterial response
- [Should be capital letter] to other biologically active compounds such as antibiotics.
- [Should be capital letter] plasmid DNA damage has been linked to the structure of verified peptidomimetics,
- [Should be capital letter] which suggests that the presence of 1,2-diarylethanoles affects the bacteria [should be point]
- LPS and generates oxidative stress, as we have already observed in our previous studies [17,18].
In our research, we observed that out of all 18 compounds used on bacterial cells, only 9 of them were toxic to them. (them twice)
Figure 8. Examples of MIC with different strains K12, R2, R3, and R4 of the studied antibiotics with.[point in the wrong place] Kanamycine, streptomycine, ciprofloxsacine, bleomycine and cloxacilline
Also here: “Due to its biological activity, combretastatin A-4 has become the subject of research both by doctors [physicians] and synthetic chemists [11-16]. Combrestatin belongs to the class of natural stilbenoids. They were isolated from the bark of the African willow Combretum caffrum in the 1980s.3 [some random number??] According to African tribes,….[according to ethnical African medicine].
Author Response
RESPOND TO REVIEWERS COMMENTS:
Thank you very much for valuable suggestions that have contributed to the improvement of the quality of work.
Reviewer 3
The authors presented interesting work in which they described the development of novel antimicrobial compounds based on the 1,2-diarylethanol scaffold.
The article is well designed and all experiments are conducted according to the good scientific level.
However, I have several issues that should be solved before the article will be accepted for the publication.
- I think that use of the word: “peptidomimetic” is misleading. Peptidomimetic is a compound which mimics the peptide structure for the better pharmacokinetic behavior of the molecule. I did not see in the paper any peptide-like structure as a precursor for the design of 1,2-diarylethanols.
We agree with the suggestion, so we changed the word peptidomimetics in the title to the word compounds. We therefore consider it more legible and concise for the recipient. We also removed all words related to peptidomimetics from the manuscript.
- The authors wrote that: “ In this study, we paid attention to the determination of the biological activity of the analyzed 1,2-diaryloethanoles as new drug substitutes…”. I think that better is to use the word: drug candidates and not “new drug substitutes”.
The wording has been corrected as suggested. The amendment is marked in green
- If authors are describing in the introduction the use of alcohols as an antiseptic agents like the basis for the development of the 1,2-diarylethanols, they should be consistence with that and they should tell that also their molecules will be used as a sprayers for the cleaning of the surfaces and hands but not as antibiotics. In this case, use of the classical antibiotics as kanamicyn for the positive control is not appropriate.
The wording has been corrected as suggested in Introduction section. The amendment is marked in green
- I did not see in the Supplemental Information all spectra images of synthesized compounds, as it is acceptable in the chemistry journals.
The NMR data are fully provided in experimental part of manuscript according to Journal requirement. The same data set was used for our recently published articles (Materials, 2020, 13, 5169, Materials, 2020, 13, 2499).
English should be improved also, by editing the paper by professional scientific English editor.
The English in the manuscript has been improved
- Also the article has a lot of typos, for example:
- Changes in the structure of the bacterial membrane and disturbances in its integrity may result in changes in the bacterial response
- [Should be capital letter] to other biologically active compounds such as antibiotics.
- ··[Should be capital letter] plasmid DNA damage has been linked to the structure of verified peptidomimetics,
- [Should be capital letter] which suggests that the presence of 1,2-diarylethanoles affects the bacteria [should be point]
- LPS and generates oxidative stress, as we have already observed in our previous studies [17,18].
All types in conclusions have been revised
In our research, we observed that out of all 18 compounds used on bacterial cells, only 9 of them were toxic to them. (them twice)
The correct form is; In our research, we observed that out of all 18 compounds used on bacterial cells, only 9 of them were toxic.
Figure 8. Examples of MIC with different strains K12, R2, R3, and R4 of the studied antibiotics with.[point in the wrong place] Kanamycine, streptomycine, ciprofloxsacine, bleomycine and cloxacilline
the signature has been corrected under the figure. The correct form is; Examples of MIC with different strains K12, R2, R3, and R4 of the studied antibiotics; kanamycine, streptomycine, ciprofloxsacine, bleomycine and cloxacilline .
Also here: “Due to its biological activity, combretastatin A-4 has become the subject of research both by doctors [physicians] and synthetic chemists [11-16]. Combrestatin belongs to the class of natural stilbenoids. They were isolated from the bark of the African willow Combretum caffrum in the 1980s.3 [some random number??] According to African tribes,….[according to ethnical African medicine].
The sentence has been redrafted as suggested by the reviewer
